# LEAN: GRAPH-BASED PRUNING FOR CONVOLUTIONAL NEURAL NETWORKS BY EXTRACTING LONGEST CHAINS

## ABSTRACT

Neural network pruning techniques can substantially reduce the computational cost of applying convolutional neural networks (CNNs). Common pruning methods determine which convolutional filters to remove by ranking the filters individually, i.e., without taking into account their interdependence. In this paper, we advocate the viewpoint that pruning should consider the interdependence between series of consecutive operators. We propose the *LongEst-chAiN* (LEAN) method that prunes CNNs by using graph-based algorithms to select relevant chains of convolutions. A CNN is interpreted as a graph, with the operator norm of each operator as distance metric for the edges. LEAN pruning iteratively extracts the highest value path from the graph to keep. In our experiments, we test LEAN pruning on several image-to-image tasks, including the well-known CamVid dataset, and a real-world X-ray CT dataset. Results indicate that LEAN pruning can result in networks with similar accuracy, while using 1.7–12x fewer convolutional filters than existing approaches.

## 1 INTRODUCTION

In recent years, convolutional neural networks (CNNs) have become state-of-the-art for many image-to-image translation tasks (LeCun et al., 2015), including image segmentation (Ronneberger et al., 2015), and denoising (Tian et al., 2020). They are increasingly used as a subcomponent of a larger system, e.g., visual odometry (Yang et al., 2020), as well as in energy-limited and real-time applications (Yang et al., 2017). In these situations, the applicability of high-accuracy CNNs may be limited by large computational resource requirements. Small networks may be more applicable in such settings, but may lack accuracy.

Neural network pruning (Mozer & Smolensky, 1989; Karnin, 1990) has recently gained popularity as a technique to reduce the size of neural networks (Blalock et al., 2020). Neural networks consist of learnable parameters, including the scalar components of the convolutional filters. When pruning, the neural network is reduced in size by removing such scalar parameters while trying to maintain high accuracy. We distinguish between individual parameter pruning (Han et al., 2016), where each parameter of an operation is ranked and pruned separately, and structured pruning (Li et al., 2017; Luo et al., 2017), where entire convolutional filters are ranked and pruned. As convolution operators can only be removed once all scalar parameters of the filter kernel have been pruned, structured pruning is favored over individual pruning when aiming to improve computational performance (Park et al., 2017). In the remainder of this paper, we focus on structured pruning.

Although structured pruning methods take into account the division of a neural network into operations, they do not take into account the fact that the output of the network is formed by a *sequence* of such operations. This has two drawbacks. First, since the relative scaling of individual convolutions may vary without changing the output of the whole chain, pruning methods that prune individual operators could potentially prune a suboptimal set of operators from the chain. Second, to significantly reduce evaluation time, a severe pruning regime must be considered, i.e., a pruning ratio (percentage of remaining parameters after pruning) of 1–10%. In this regime, pruning can result in network disjointness, i.e., the network contains sections that are not part of some path from the input to the network output. Some existing pruning methods take into account network structure to

a limited degree (Salehinejad & Valaee, 2021). In practice, however, these methods do not contain safeguards to avoid network disjointness.

In this paper, we present a novel pruning method called LongEst-chAiN (LEAN) pruning, which as opposed to conventional pruning approaches uses graph-based algorithms to keep or prune chains of operations collectively. In LEAN, a CNN is represented as a graph that contains all the CNN operators, with the operator norm of each operator as edge weights. We argue that strong subnetworks in a CNN can be discovered by extracting the longest (multiplicative) paths, using computationally efficient graph algorithms. The main focus of this work is to show how LEAN pruning can significantly improve the computation speed of CNNs for real-world image-to-image applications, and obtain high accuracy in the severe pruning regime that is difficult to achieve with existing approaches.

This paper is structured as follows. In Section 2, we explore existing pruning approaches. In Section 3, we outline the preliminaries on CNNs, pruning filters, and the operator norm. Next, in Section 4, we introduce LEAN pruning and describe how to calculate the operator norm of various convolutional operators. We discuss the setup of our experiments in Section 5. In Section 6, we demonstrate the results of the proposed pruning approach on a series of image segmentation problems and report practically realized wall time speedup. Our final conclusions are presented in Section 7.

## 2 RELATED WORK

Reducing the size of neural networks by removing parameters has been studied for decades (Mozer & Smolensky, 1989; Karnin, 1990; Hassibi et al., 1993). Several works take into account the structure of the network to some degree. In Lin et al. (2017) filters are pruned at runtime based on the feature maps (Lin et al., 2017). Alternatively, one can prune entire channels (He et al., 2017), or decide which channels to keep so that the feature maps approximate the output of the unpruned network over several training examples (Luo et al., 2017). In recent work, a graph is built for each convolutional layer, and filters are pruned based on the properties of this graph (Wang et al., 2021). In Salehinejad & Valaee (2021) a neural network is represented as a graph and interdependencies are determined using the Ising model.

Many pruning approaches are aimed at reducing neural network size with little accuracy drop (Dong & Yang, 2019; He et al., 2019; Molchanov et al., 2019; Zhao et al., 2019), as opposed to sacrificing accuracy in favor of computation speed. These approaches rarely exceed a pruning ratio of 12–50% (Blalock et al., 2020; Luo et al., 2017; Lin et al., 2019). When a high pruning ratio is used, e.g., a range of 5–10% (Lin et al., 2017; Liu et al., 2019), a significant drop in accuracy is observed. Pruning ratios of 2–10% can be achieved with an accuracy drop of 1-3% by learning-rate rewinding (Renda et al., 2020). However, the reduction in FLOPs was less substantial (1.5–4.8 times). In Yeom et al. (2021) severe pruning ratios of up to 1% have been considered, but the approach achieved limited improvements in terms of FLOPs reduction compared with existing pruning methods.

Criteria for deciding which elements of a neural network to prune have been extensively studied. A parameter's importance is commonly scored using its absolute value. Whether this is a reasonable metric has been questioned (LeCun et al., 1990). Singular values (which determine certain operator norms) have been used to compress network layers (Denton et al., 2014) and to prune feed-forward networks (Abid et al., 2002). Efficient methods for the computation of singular values have been developed for convolutional layers (Sedghi et al., 2019). Furthermore, a definition of ReLU singular values was proposed recently with an accompanying upper bound (Dittmer et al., 2019).

## 3 PRELIMINARIES

### 3.1 CNNS FOR SEGMENTATION

A common image to image translation task is semantic image segmentation. The goal of semantic image segmentation is to assign a class label to each pixel in an image. A segmentation CNN computes a function $f : \mathbb{R}^{m \times n} \to [0, 1]^{k \times m \times n}$, which specifies the probability of each pixel being in one of the $k$ classes for an $m \times n$ image.

CNNs are composed of layers of operations which pass images from one layer to the next. Every operation, e.g., convolution, has an input $\mathbf{x}$ and output $\mathbf{y}$. The input and output consist of one

or more images, called channels. For clarity, we distinguish throughout this paper between an **operation**, which may have several input and output channels, and an **operator**, which computes the relation between a single input channel and a single output channel. For instance, in a convolutional operation with input channels $\mathbf{x}_1, \ldots, \mathbf{x}_N$, an output channel $\mathbf{y}_j$ is computed by convolving input images with learned filters

$$\mathbf{y}_j = \left( \sum_{i=1}^{N} h_{ij} * \mathbf{x}_i \right) + b_j. \tag{1}$$

Here $h_{ij}$ is the filter related to the convolution operator that acts between channel $\mathbf{x_i}$ and $\mathbf{y_j}$, and $b_j$ is an additive bias parameter. In a similar way, every CNN operation produces an output which consists of a number of channels. The exact arrangement of operations, and connections between them, depends on the architecture.

A common operator to downsample images is the strided convolution. The stride defines the step size of the convolution kernel. A convolution with stride $s$ defines a map $h : \mathbb{R}^{m \times n} \to \mathbb{R}^{\frac{m}{s} \times \frac{n}{s}}$. Upsampling images can be done by transposed convolutions. Transposed convolutions intersperse the input image pixels with zeroes so that the output image has larger dimensions.

In addition to convolution operators, other common operators such as pooling and batch normalization are often used. A batch normalization operator (Ioffe & Szegedy, 2015) normalizes the input images for convolutional layers. A batch normalization operator scales and shifts an image $\mathbf{x}_i$ by

$$\mathbf{y}_i = \gamma \frac{\mathbf{x}_i - \mu_B}{\sqrt{\sigma_B^2 + \epsilon}} + \beta. \tag{2}$$

Here, $\gamma$ and $\beta$ are scaling and bias parameters which are learned during training, and $\mu_B$ and $\sigma_B^2$ are the running mean and variance of the mini-batch, i.e., the set of images used for the current training step. For an overview of CNN components we refer to Goodfellow et al. (2016).

### 3.2 PRUNING CONVOLUTION FILTERS

Pruning techniques aim to remove extraneous parameters from a neural network. Several schemes exist to prune parameters from a network, but retraining the network after pruning is critical to avoid significantly impacting accuracy (Han et al., 2015). Pruning a network once after training is called one-shot pruning. Alternatively, a network can be fine-tuned, where the network is repeatedly pruned by a certain percentage and is retrained for a few epochs after every pruning step. Fine-tuning typically gives better results than one-shot pruning (Renda et al., 2020).

**Generic pruning algorithm:** All pruning methods used in this work make use of the fine-tuning pruning algorithm outlined in Algorithm 1. The selection criteria for determining which filters to keep for each step define the different pruning methods. The pruning ratio `pRatio` is the fraction of remaining convolutions we ultimately want to keep, and `stepRatio` is the fraction of convolutions that is pruned at each step.

---

**Algorithm 1** Fine-tuning pruning algorithm

---

1: **procedure** PRUNE(MODEL, PRATIO, NSTEPS, EPOCHS)
2:     stepRatio $\leftarrow e^{\ln(pRatio)/nSteps}$
3:     **for** step $\leftarrow$ 0 to $nSteps$ **do**
4:         pruneParams $\leftarrow$ selectPrunePars(model, stepRatio)
5:         model $\leftarrow$ removePars(model, pruneParams)
6:         **for** k $\leftarrow$ 0 to $epochs$ **do**
7:             model $\leftarrow$ trainOneEpoch(model, trainData)
        **return** model

---

Here, we focus on structured pruning. In structured pruning, a common approach to decide which filters to remove is *structured magnitude pruning*. When using structured magnitude pruning, a convolution filter $\mathbf{h} \in \mathbb{R}^{k \times k}$ is scored by its $L_1$ vector norm $||\mathbf{h}||_1$. Filters with norms below a threshold are pruned. The threshold is determined by sorting a group of filters, and removing a percentage based on the pruning ratio. Thresholds can be set per layer or globally. Setting thresholds globally can give higher accuracy than setting thresholds per layer (Blalock et al., 2020).

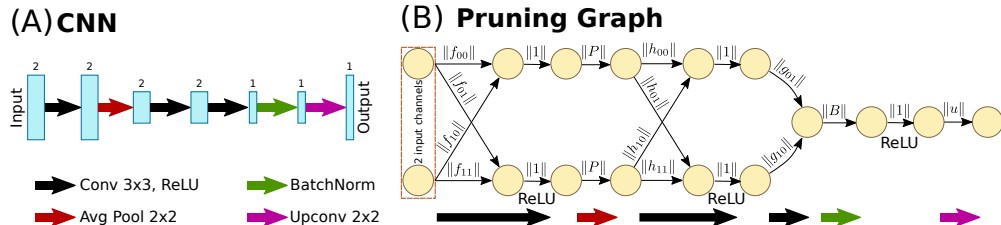

Figure 1: (A) Example CNN architecture with the number of channels indicated above each layer. (B) Associated pruning graph. Every channel is a node, and every operator is an edge connecting input and output nodes. The edge weights are the corresponding operator norms.

### 3.3 OPERATOR NORM

As an alternative to the $L_1$-norm, one can interpret a convolution $h$ as a linear operator which acts on the input image, and score it according to an operator norm. The (induced) operator norm $\|\cdot\|_p$ is defined as

$$\|h\|_p := \sup \left\{ \|h * \mathbf{x}\|_p \ \Big| \ \mathbf{x} \in \mathbb{R}^n, \|\mathbf{x}\|_p = 1 \right\}. \tag{3}$$

A common operator norm is the *spectral norm*, which is induced by the Euclidean norm ($p = 2$). The spectral norm can be obtained by calculating the largest singular value of the matrix $H$ associated with $h$, $\|h\|_2 = \sigma_{max}(H)$ (Meyer, 2000). A property that we will use is that the spectral norm is submultiplicative, i.e., we have

$$\|AB\| \le \|A\| \cdot \|B\|, \quad \text{for all } A, B \in \mathbb{R}^{n \times n}. \tag{4}$$

## 4 METHOD

The idea behind LEAN is to construct a weighted graph structure formed by the operators of the CNN and having their respective norms as edge weights, such that the multiplicative longest paths in this graph are selected as important subnetworks. The remaining unselected operators will then be pruned. The motivation for LEAN is two-fold. The first consideration is that since convolutions are linear operators, the scaling of an individual convolution within a chain of convolutions is somewhat arbitrary. For a scalar $\lambda$ and chain of linear operators $A_1 \circ \cdots \circ A_m$, we have that any chain $(\lambda_1 A_1) \circ \cdots \circ (\lambda_m A_m)$ is equivalent if $\prod \lambda_i = 1$. Since $\|\lambda_i A_i\| = |\lambda_i| \|A_i\|$, this can lead to any arbitrary ranking of operators. However, the chain as a whole gives the same output for each input. We argue that this can lead to incorrect pruning when pruning individual operators based on norms, rather than entire chains. We hypothesize that these scaling properties still approximately hold in the presence of non-linear operators. Since LEAN ranks chains of operators, it is invariant under these scaling properties. Second, by extracting chains LEAN combats network disjointness.

### 4.1 LEAN: CREATING THE PRUNING GRAPH

**Graph structure:** In this section, we define the pruning graph that is the basis of the LEAN algorithm. As discussed in Section 3.1, we say that every CNN operation has an input $\mathbf{x}$ and an output $\mathbf{y}$, consisting of channels $\mathbf{x}_i$ and $\mathbf{y}_i$. For each channel, we add a single node in the pruning graph. An edge connects two nodes corresponding to input channel $\mathbf{x}_i$ and output channel $\mathbf{y}_i$ if channel $\mathbf{x}_i$ is used in the computation of channel $\mathbf{y}_i$. In the terminology of Section 3.1, each edge corresponds to an operator. For instance, a convolution operation is converted by adding an edge from each input channel $\mathbf{x}_i$ to every output channel $\mathbf{y}_j$, each corresponding to exactly one filter $h_{ij}$. Certain CNN operations may be performed in-place in practice, but we consider them as separate nodes in the pruning graph. A combined convolution and ReLU operation, for instance, results in a node for the output of the convolution and a separate node for the output of the ReLU, as shown in Figure 1.

**Edge weights:** To each edge, we assign as weight the operator norm of its corresponding operator. That is, we calculate the maximal scaling that an input could undergo as a result of applying the operator. In this calculation, we ignore any additive bias terms. For instance, applying a batch normalization results in a scaling of $|\gamma|/\sqrt{\sigma_B^2 + \epsilon}$ (see Equation (2)). The scaling of non-linear operators

in neural networks is sometimes bounded, as in the case of the ReLU for instance, to which we assign a weight of 1. We describe the calculation of operator norms for various convolution operators in Section 4.3.

**Path lengths:** The length of a path in the graph is determined by multiplying the edge weights. LEAN aims to model the norm of the composition of the operators corresponding to the edges. Equation (4) states that $\|A\| \cdot \|B\|$ is an upper bound for $\|AB\|$. For LEAN, we assume that $\|AB\| \approx \|A\| \cdot \|B\|$ is a reasonable approximation, although it may not hold generally. By defining the path length as the multiplication of the edge weights (operator norms), path lengths are invariant under scaling linear operators in a chain if the scalars multiply to 1.

There are some edge cases to consider. First, some CNNs contain operations that are meant to distribute features throughout the network, but are not implemented with learnable parameters, e.g., residual connections in ResNet (He et al., 2016). We include residual connections in the pruning graph with an edge weight of 1, but label them as unprunable to prevent the residual connections from being removed from the network. Second, we do not consider CNNs with recurrent connections. Therefore, the pruning graph is a Directed Acyclic Graph (DAG). Large pruning graphs can be reduced in size, e.g., by merging nodes that are connected by a single edge (see Appendix B).

## 4.2 LEAN: EXTRACTING CHAINS FROM THE GRAPH

The LEAN method prunes chains of convolutions based on paths in the graph; we **keep** the longest paths (with the highest multiplicative operator norm). We refer to this as LongEst-chAiN (LEAN) pruning. When we perform LEAN pruning, we iteratively extract such paths from the graph until we have reached the pruning ratio. This means that the edges that are not extracted are pruned. Finding paths is done by iteratively running an all-pairs longest path algorithm Cormen et al. (2009).

---

**Algorithm 2** LEAN

```
1: procedure LEAN(MODEL, PRUNERATIO)
2:     graph ← createPruningGraph(model)
3:     retainedConvs ← []
4:     while fractionRemainingConvs < 1 − pruneRatio do
5:         bestChain ← longestPath(graph)
6:         retainedConvs ← retainedConvs + bestChain
7:         graph ← removeFromGraph(graph, bestChain)
8:     convsToPrune ← convsInModel − retainedConvs
9:     return convsToPrune
```

---

LEAN pruning is incorporated in the fine-tuning procedure. For each pruning step in the fine-tuning procedure, Algorithm 2 is used to select the filters to prune (line 4 in Algorithm 1). For DAGs, the longest path in a graph can be found in $\mathcal{O}(|V| + |E|)$ time, where $V$ is the set of nodes, and $E$ is the set of edges (Sedgewick & Wayne, 2011). For a CNN with $m$ channels (nodes), and $k$ operators (edges), we can extract a longest path in $\mathcal{O}(m + k)$ time. As we extract at least one operator with every execution of line 5 in Algorithm 2, the longest path algorithm is run at most $k(1-p)$ times. So the worst-case complexity of Algorithm 2 is $\mathcal{O}(k(1-p)(m+k))$ for a pruning ratio $p$. Unprunable edges can be part of a longest path to extract operators, but do not count towards the pruning ratio.

After every LEAN pruning step is concluded, some post-processing is performed. In some cases there are channels which receive no input data at all, or which are equal to a homogeneous constant image for all input data. We therefore remove nodes without incoming edges as well as nodes where the succeeding batch normalization has running variance below some threshold ($10^{-40}$ by default). A low running variance can occur when the output of a convolution is always zero after applying the ReLU activation function, for instance. Second, bias terms are removed from the CNN when all associated convolution or batch normalization operations are pruned.

## 4.3 OPERATOR NORM CALCULATION

**Operator norm of convolutions**: For a convolution filter $h$ and an $n \times n$ image, $h^+$ is the filter padded with zeros to size $n \times n$. The singular values of $h$ are the magnitudes of the complex entries

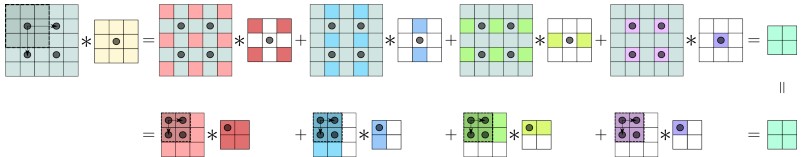

Figure 2: The output of a stride-2 convolution can be obtained by adding the result of 4 convolutions. Split the image and filter into the coloured sections, with white entries representing zeroes, and sum the outputs pixel-wise. The dots in the image represent the positions of the center of the filter as it moves over the image.

of the 2D Fourier transform $F_{2D}(h^+)$ (Section 5 of Jain (1989))

$$\sigma_{max}(H) = \max\left\{\left|F_{2D}(h^+)\right|\right\} \qquad (5)$$

**Downsampling: operator norm for strided convolutions**: A strided convolution is equal to the sum of regular convolutions on smaller input images (see Figure 2). A single parameter of a stride-2 convolution filter is multiplied only with every other pixel (horizontally and vertically). Similarly, filter parameters which are 2 apart are multiplied with the same pixels. Here, we calculate the operator norm for a stride-2 convolution operator $h$. Let $h^{[i]}$ and $X^{[i]}$ be the partitioned convolution kernels and input images, both zero-padded to the correct size. For a stride-2 convolution we have

$$h *_2 X = \sum_{i=1}^{4} h^{[i]} * X^{[i]}. \qquad (6)$$

We can apply Equation 5 to obtain the singular values of $h^{[i]}$. Equation 6 is analogous to the equation of a convolutional layer with 4 input channels, and 1 output channel. The operator norm of a convolutional layer can be computed by means of a tensor $P \in \mathbb{R}^{4 \times 1 \times n \times n}$ (Sedghi et al., 2019)

$$P_{c_{in},c_{out},i,j} = F_{2D}(h^{[c_{in}]+})_{i,j}.$$

According to Theorem 6 of Sedghi et al. (2019), the spectral norm of the convolutional layer equals the maximum of the singular values of the $4 \times 1$ matrices $P_{:,:,i,j}$. Since the matrices are single-column, their singular value equals their $L_2$-norm. Therefore, the spectral norm of $h$ equals

$$\|h\| = \max_{i,j}\left\{\sum_{c_{in}} P^2_{c_{in},:,i,j}\right\}. \qquad (7)$$

**Upsampling: operator norm for transposed convolutions**: The matrix of a stride-$s$ transposed convolution is the transposed matrix of a stride-$s$ convolution (Long et al., 2015). Since we have $\|A\| = \|A^T\|$, for a transposed convolution $h$, the operator norm can be computed by Equation 7.

## 5 EXPERIMENTAL SETUP

In our experiments, we compare LEAN pruning to several structured pruning methods across three image segmentation datasets and three CNN architectures: MS-D, U-Net4, and ResNet50. We assess the performance of pruned neural networks across 5 independent runs of fine-tuning (Algorithm 1). For each dataset, we have trained a single model as a starting point for pruning. In every experimental run, the same trained model was pruned.

For all pruning methods, we measure the pruning ratio as the fraction of convolutions remaining: $\sum_{h \in H} M(h)/|H|$ where $H$ is the set of all convolutions in a network, and $M(h)$ is 0 if a convolution $h \in H$ is pruned and 1 otherwise. This means that other parameters, such as batch normalization and bias parameters, are pruned when the associated convolutions are pruned, but do not count towards the pruning ratio.

The MS-D networks were pruned to a pruning ratio of 1% in 45 steps. The U-Net4, and ResNet50 networks were pruned to a ratio of 0.1% in 30 steps. All were retrained for 5 epochs after each step. We chose relatively severe pruning ratios because we are interested in significant computational speedup. U-Net4 and ResNet50 are pruned to a lower ratio as they have orders of magnitude more convolutions than the MS-D network.

### 5.1 STRUCTURED PRUNING METHODS

We compare LEAN to two layer-wise pruning methods, and two global pruning methods. The layer-wise pruning methods are geometric median pruning (**GM**) (He et al., 2019), and soft filter pruning (**SFP**) (He et al., 2018). The first global pruning method we compare to is **structured magnitude pruning**, i.e., pruning entire filters by their $L_1$-norm (see Section 3.2), and the second global method we compare to is **operator norm pruning**, i.e., pruning entire filters using the operator norm. Here, we choose to use the spectral norm. The difference between LEAN and structured magnitude pruning is 1) the operator norm; 2) the consideration of network structure. The comparison of LEAN to structured operator norm pruning measures the effect of incorporating the network structure.

We compare LEAN to structured magnitude pruning and operator norm pruning for all CNN architectures. Both GM and SFP contain implementations to prune ResNet50 but not MS-D or U-Net4, and hence are used only in the ResNet50 experiments.

### 5.2 CNN ARCHITECTURES

In this section, we describe three fully-convolutional CNN architectures that are used in the experiments: the Mixed-Scale Dense convolutional neural network (MS-D) network (Pelt & Sethian, 2018), U-Net (Ronneberger et al., 2015), and ResNet (He et al., 2016). Table 1 outlines which operators are present in the networks. The networks were trained using ADAM (Kingma & Ba, 2014) with $lr = 0.001$, minimizing the negative log likelihood function.

| CNN | Convolution | | | Pooling | Batch normalization | # Parameters | Edges in pruning graph |
| | Strided | Transposed | Dilated | | | | |
|---|---|---|---|---|---|---|---|
| MS-D | No | No | Yes | No | No | $4.57 \cdot 10^4$ | $5.05 \cdot 10^3$ |
| ResNet50 | Yes | No | Yes | Yes | Yes | $3.29 \cdot 10^7$ | $1.44 \cdot 10^7$ |
| U-Net4 | No | Yes | No | Yes | Yes | $1.48 \cdot 10^7$ | $1.84 \cdot 10^6$ |

Table 1: Operators present in MS-D, ResNet50, U-Net4 architectures.

In our experiments we use MS-D networks as described in Pelt & Sethian (2018) and implemented in Hendriksen (2020). Every layer has 1 channel and all convolutions have a dilated $3 \times 3$ filter, except the final $1 \times 1$-layer. The dilations for layer $i$ were set to $1 + (i \mod 10)$. The final layer is excluded from pruning as it contributes less than 0.5% of FLOPs.

As U-Net architecture we use a fully-convolutional (FCN) U-Net4 network, i.e., a U-Net with 4 scaling operations. We used a U-Net4 architecture from the PyTorch-UNet repository Milesi (2020). As ResNet architecture, we use an FCN-ResNet50 network (He et al., 2016). The ResNet50 model is adapted from PyTorch's model zoo code. We replace the max pooling layers of ResNet and U-Net with average pooling layers as average pooling is a linear operator which can be modeled as a strided convolution for which we can compute the operator norm. In some cases U-Net with average pooling can perform better than with max pooling (Astono et al., 2020).

### 5.3 DATASETS

In our experiments, we consider three datasets: a high-noise, but relatively simple, segmentation dataset (Pelt & Sethian, 2018) (CS dataset); the well-known CamVid dataset (Brostow et al., 2008; 2009); and a real-world X-ray CT dataset (Coban & Lucka, 2019; Coban et al., 2020) to test the methods in practice. The CS dataset is a 5-class segmentation dataset of 1000 training, 250 validation, and 100 test images. As a starting point for pruning, we trained a 100-layer MS-D network with an accuracy of 97.5%, ResNet50 with 95.8% accuracy, and U-Net4 with 97.6% accuracy.

The X-ray CT dataset consists of 9216 training, 2048 validation, and 1536 test images. As in Schoonhoven et al. (2020), we use the F1-score to quantify results. As a starting point for pruning, we trained a 100-layer MS-D network with a 0.88 F1-score, ResNet50 with a 0.85 F1-score, and U-Net4 with a 0.88 F1-score. As in Paszke et al. (2017), experimental results on the CamVid dataset are quantified using mean Intersection-over-Union (mIoU). As a starting point for pruning, we trained a 150-layer MS-D with 0.52 mIoU, ResNet50 with 0.71 mIoU, and U-Net4 with 0.65 mIoU. More details on the datasets can be found in Appendix A.

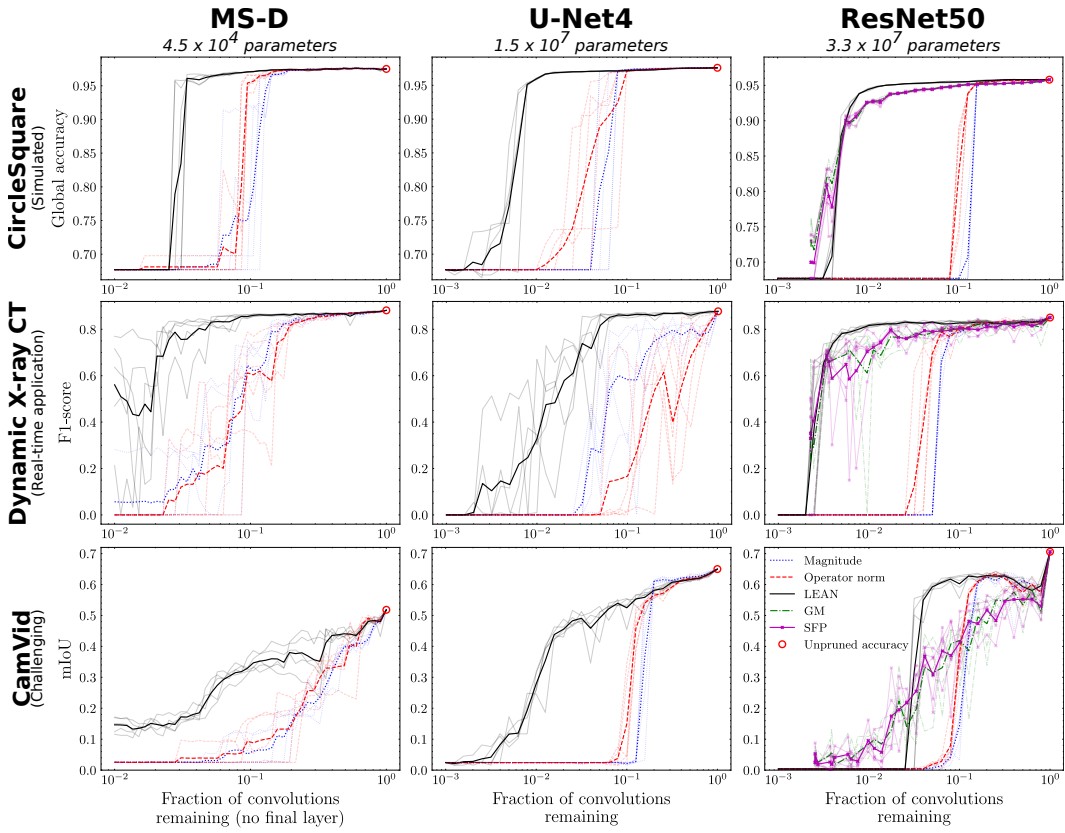

Figure 3: Comparison of structured pruning methods and LEAN pruning on three datasets (rows). Pruning methods are applied to MS-D, U-Net4, ResNet50 network architectures. The base model is pruned to a ratio of 1% (MS-D) or 0.1% (ResNet50, U-Net4) for all datasets. This is repeated five times (translucent lines) and the average is taken (solid lines).

## 6 RESULTS

### 6.1 EXPERIMENTAL RESULTS FOR SEVERE PRUNING

The results of the pruning experiments are displayed in Figure 3, showing that LEAN pruning at similar accuracy obtains networks with a lower pruning ratio than the four compared methods. The pruning ratio that can be achieved without significant loss of accuracy depends on the network architecture and the complexity of the dataset.

In the CS dataset results, we notice a drop-off point where performance decreased significantly for all networks. On average over 5 runs of pruning, LEAN achieved a 3.4%, 0.79%, and 0.79% pruning ratio for MS-D, U-Net4, and ResNet50, with an average accuracy reduction of 1.4%, 2.5%, and 2.1% respectively. On ResNet50, at an accuracy reduction of 3% compared to the unpruned network, LEAN achieves a 43% reduction in the number of convolutions compared to GM and SFP. Below 15% accuracy reduction SFP and GM perform better, but the network no longer reliably segments the data at these accuracies.

On the dynamic X-ray CT dataset, we notice large fluctuations in F1 for MS-D and U-Net4. This may be due to the F1-score which is defined as a reciprocal. In addition, on U-Net4, LEAN performs better than the structured pruning methods right from the start. On average over 5 runs, LEAN achieved a 5.1% and 6.3% pruning ratio for MS-D and U-Net4, with an average F1 drop of 5.7%, and 3.3% respectively. For ResNet50, a drop-off point is again observed, which occurs at a significantly lower pruning ratio for LEAN than for both global pruning methods. GM and SFP exhibit a more gradual reduction in F1-score.At an F1-score reduction of 3% compared to the unpruned network, LEAN achieves a 92% reduction in the number of convolutions compared to GM and SFP.

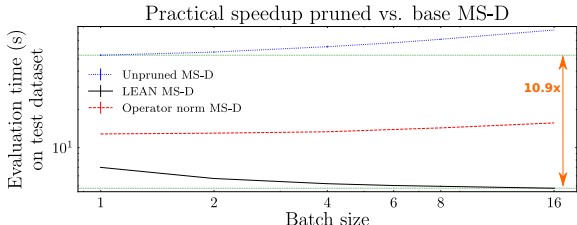

Figure 4: Practically realized speedup of pruned MS-D networks evaluated on the test dataset.

On the CamVid dataset, we observe a declining mIoU for MS-D and U-Net4 as pruning progresses on this challenging dataset. On ResNet50 we notice an initial drop in mIoU, but subsequent pruning steps increase the performance initially. These observations could indicate that 5 epochs of retraining are not sufficient for the CamVid dataset to recover performance. Interestingly for U-Net4, we notice that for two pruning ratios, structured magnitude pruning appears to perform slightly better than LEAN. Given the variance between different runs, possibly due to the limited number of retraining epochs, we suspect that this difference is not statistically significant. After the initial drop in mIoU, we notice a later drop-off pruning ratio on ResNet50. LEAN achieves a 6.3% pruning ratio with an average mIoU reduction of 14.3% on ResNet50, whereas the best performing other method dropped-off at a 20.0% pruning ratio. Interestingly, both layer-wise pruning methods GM and SFP exhibit a sustained reduction in test mIoU rather than the drop-off we observe for the global pruning methods.

### 6.2 Speedup real-world dynamic X-ray CT segmentation

In addition to measuring the achievable pruning ratios, we measure the practically realized wall-time speedup. We tested this on the dynamic X-ray CT dataset for which a speedup has immediate benefits in practice. Existing pruning support in PyTorch only masks pruned filters, thereby not creating a faster network. Therefore, we implemented a custom MS-D model which loads only the unpruned filters. During the experiments, an MS-D network pruned to a ratio of 2.5% (40-fold reduction) with LEAN achieved an F1-score of 0.83 (drop of 5.4%). This network was 10.9 times faster than the unpruned network in practice. The speed of evaluating the entire test set is impacted by the batch size, which we take into account as shown in Figure 4. For comparison, we included the best performing pruned MS-D network by an other pruning method (operator norm pruning) which achieved a pruning ratio of 15.8% with an F1-score of 0.83. We show differences between MS-D networks pruned with different pruning methods in Appendix C.

## 7 Conclusion

In this paper, we have introduced a novel pruning method (LEAN) for CNN pruning by extracting the highest value paths of operators. We incorporate existing graph algorithms and computationally efficient methods for determining the operator norm. We show that LEAN pruning permits removing significantly more operators while retaining better network accuracy than several existing pruning methods. Our results show that LEAN pruning can increase the speed of the network, both in theoretical speedup (FLOPs reduction) and in practice. In conclusion, LEAN enables severe pruning of CNNs while maintaining a high accuracy, by effectively exploiting the interdependency of network operations.

Future work could be split along several lines. First, there are more CNN operators for which methods to compute their operator norms could be developed. Notably, we have mostly disregarded non-linear operators in this work. Next, LEAN approximates the norm of a chain of operators using the submultiplicative property upper bound $\|AB\| \leq \|A\| \cdot \|B\|$. New methods for approximating the norm of a chain of composed operators could strengthen LEAN as it more accurately extracts chains with strong operator norms. In addition, new graph theoretic approaches for extracting meaningful paths from the graph could be explored. Such algorithms are already abundant in the field of graph theory, and could quite readily be carried over to neural network pruning research. Lastly, LEAN currently works by greedily extracting high-value paths. Approaches such as Guo et al. (2016) could be considered to avoid greedy selection of operators.

## 8 REPRODUCIBILITY STATEMENT

In order to aid reproducibility the authors have published (Python) code for LEAN pruning. There are installation instructions that aim to help users install and run the code on their machines, as well as explanations and documentation for the code and scripts. The scripts are intended to be usable on relatively simple machines (with GPU and CUDA), and to be limited to several minutes run time. First, the code contains a script to generate the CircleSquare (CS) dataset. Second, the code contains a script to train an MS-D network (a pre-trained network is also supplied) on the CS dataset. The script uses the MS-D network as it is the least computationally expensive to run, but the U-Net4 and ResNet50 models and pruning methods are also supplied. Third, the code contains a script to run LEAN pruning, and the two global pruning methods, on the trained MS-D network for the CS dataset (example results are also supplied). This way, users can check the experimental results, and experiment themselves with the LEAN method. In addition, the code contains several testing procedures that aim to verify the correctness of the function in the code. For example, the Fourier-based method of computing the operator norm is tested against the power method for random convolutions and strides, and against explicitly computing the SVD of the matrix that is associated with a convolution. Another example is that these tests check whether the pruning methods actually prune the correct number of convolutions. The code is submitted as .zip file in the supplementary materials section, with README instructions.

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

## A APPENDIX: DATASETS

In this appendix we discuss some more details on the datasets used for experimentation.

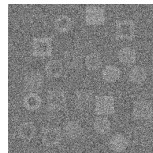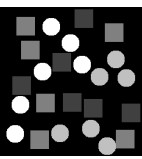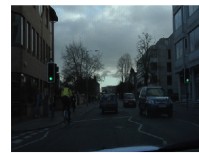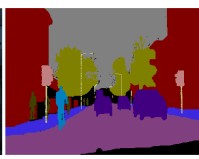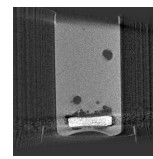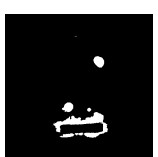

Figure 5: Example input and target images of the (Left) Circle-Square (CS), (middle) CamVid, (right) real-world dynamic CT datasets.

**Simulated Circle-Square (CS) dataset:** We used a simulated high-noise 5-class segmentation dataset containing $256 \times 256$ images of randomly placed squares and circles (CS dataset) (Pelt & Sethian, 2018) (see Figure 5). The objects were assigned a random grey value and Gaussian noise was added to the images. In total, we generated 1000 training, 250 validation, and 100 test images. Experimental results on the CS dataset are quantified using global accuracy, i.e., the ratio of correctly classified pixels, regardless of class, to the total number of pixels.

**CamVid:** The Cambridge-driving Labeled Video Database (CamVid) (Brostow et al., 2008; 2009) is a collection of videos with labels, captured from the perspective of a driving automobile. In total, 700 labeled frames are split into 367 training, 100 validation, and 233 test images. As there are few training images, we combined the training and validation datasets and trained for a fixed 500 epochs. Similar to other papers that apply CNNs to CamVid (Badrinarayanan et al., 2017; Paszke et al., 2017), we use 11 classes, and a single class representing unlabeled pixels (see Figure 5).

We used median frequency balancing (Eigen & Fergus, 2015) to balance classes for training, and set the unlabeled class weights to zero. During training, we used data augmentation by cropping and (horizontally) flipping input images.

**Real-world dynamic CT dataset:** The real-time dynamic X-ray CT dataset contains images of a dissolving tablet suspended in gel (Coban & Lucka, 2019; Coban et al., 2020). The bubbles are to be segmented within a glass container filled with gel (Schoonhoven et al., 2020) (see Figure 5). The dataset consists of $512 \times 512$ images, split into 9216 training images, 2048 validation images, and 1536 test images. As in Schoonhoven et al. (2020), we use the F1-score because the large amount of background pixels make global accuracy an unsuitable metric.

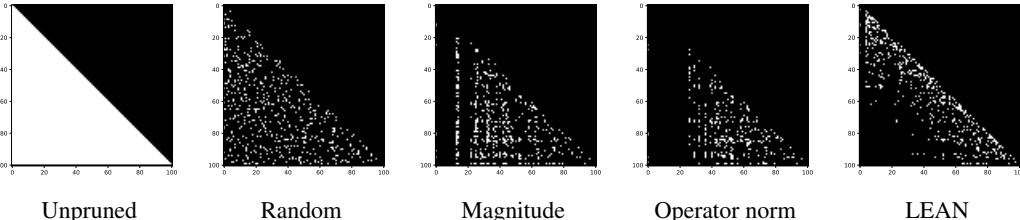

**Figure 6:** Adjacency matrices of active convolutions (in white) after pruning. All pruned network were pruned to a ratio of 10%. From left to right, we have the unpruned matrix of a 100-layer MS-D network trained on the real-world dynamic CT dataset, randomly pruned convolutions, structured magnitude pruning, structured operator norm pruning, and LEAN.

## B   APPENDIX: REDUCING THE SIZE OF THE PRUNING GRAPH

The procedure outlined in Section 4 can lead to large pruning graphs, but the size of the graph can be reduced. First, according to Equation 3, the operator norm of ReLU is 1. Therefore, the combination of a convolution followed by a ReLU can be combined into a single edge whose weight equals the norm of the convolution.

Batch normalization often succeeds a convolution. Batch-normalization scaling is applied with different learned parameters per input channel, and output a single channel. Therefore, the input convolution edge and the following batch normalization edges can be combined. The edges can be combined into a single edge whose weight is the product of the two edge weights, preserving the path length.

## C   APPENDIX: STRUCTURE OF PRUNED MS-D NETWORKS

To investigate the structure of pruned networks we plotted the adjacency matrices of pruned networks where an entry is 0 if it is pruned (black) and 1 if it is still active (white). Here, we show the adjacency matrices of MS-D networks pruned to a ratio of 10% in Figure 6. After pruning, LEAN retains only connections linked to nearby layers in the densely connected MS-D network. Compared to individual filter pruning, LEAN exposes a distinct structure which may suggest that LEAN could be used for architecture discovery.

