# OpenReview forum: "LEAN: graph-based pruning for convolutional neural networks by extracting longest chains"
_ICLR.cc/2022/Conference — ICLR 2022 Submitted_

### Official Review · Reviewer_ami3 · 2021-10-31

**Correctness:** 3
**Technical Novelty And Significance:** 3
**Empirical Novelty And Significance:** 2
**Recommendation:** 5
**Confidence:** 3

**Main Review:**

The idea is somewhat novel and sound. It is proved to be effective in the evaluation.

**Summary Of The Paper:**

This paper proposes a pruning method for CNNs by using a graph-based algorithm. It was evaluated by using CS, CamVid, and dynamic CT dataset.

**Summary Of The Review:**

The size of the databases used for the evaluation seems rather small. Hence, the improvement may be obtained from the adaptation of models to the target domain. It would be better to evaluate this method by using datasets with a larger size to clarify this point. Also the comparison with the existing pruning methods are not sufficient.

---

> ### Comment · Area_Chair_8iJk · 2021-11-10
> **Low-Quality Review**
>
> Dear Authors -
>
> This is a low-quality review. It provides neither useful feedback nor concrete comments for you to respond to. I have requested an update from the reviewer, but I have not heard back.
>
> Please make your best effort to respond to this review. However, know that this review will not be an important consideration in my metareview.
>
> Sincerely,
> Your AC

---

> ### Author Response · Authors · 2021-11-22
> **Response to Reviewer ami3**
>
> We thank the reviewer for the useful feedback, which has resulted in several significant improvements in the manuscript. As mentioned in the general update, we agree with the comment made by several reviewers that more comparisons to state-of-the-art pruning methods are necessary. We performed experiments on ResNet50 for all 3 datasets for geometric median pruning (GM) [1], and soft filter pruning (SFP) [2] and have added the results to the manuscript. Furthermore, we have tried to improve the manuscript using the reviewer’s feedback, as follows:
>
> 1. ***The size of the databases used for the evaluation seems rather small. Hence, the improvement may be obtained from the adaptation of models to the target domain. It would be better to evaluate this method by using datasets with a larger size to clarify this point.***
>
> We understand the reviewers point related to standard classification benchmarks such as ImageNet. In our work we aim to provide significant practical speedup for imaging tasks in real-world applications. These real-world datasets are often small, yet accelerating CNNs has practical benefits in these scenarios. Imaging tasks such as the X-ray CT segmentation task often have very little training data available, for example because they involve manual annotation by domain experts, yet the trained segmentation CNN is applied in real time. Instead of adding ImageNet benchmarks, we think the weaknesses raised by the reviewer are alleviated by comparing to several more advanced methods on our current datasets, which we have included in the revised manuscript. We do aim to benchmark LEAN on the ImageNet database in the future.
>
> In terms of adaptation, we would like to ask the reviewer to clarify the point. We have split the data into training, validation and test sets to mitigate overfitting on the training data.
>
> 2. ***Also the comparison with the existing pruning methods are not sufficient.***
>
> We agree with the reviewer’s remarks and we added two experimental comparisons to more advanced methods. We have added experimental results for GM [1] and SFP [2] for ResNet50 for all datasets.
>
> We would once again like to thank the reviewer for the feedback. We hope we have answered your questions.
>
> [1] Filter Pruning via Geometric Median for Deep Convolutional Neural Networks Acceleration, CVPR 2019
>
> [2] Soft Filter Pruning for Accelerating Deep Convolutional Neural Networks, IJCAI 2018

---

### Official Review · Reviewer_xFHB · 2021-10-31

**Correctness:** 3
**Technical Novelty And Significance:** 2
**Empirical Novelty And Significance:** 2
**Recommendation:** 5
**Confidence:** 4

**Main Review:**

Pros:
1. The proposed approach Use graph-based concepts for network pruning, which is a new approach that raises researchers' interest recently.
2. The proposed approach can prune the network with large pruning ratios.
3. The proposed approach works on segmentation tasks.

Cons:
1. The introduction and related work sections are not described well.

    The motivation of the proposed methods is not described well. Avoiding disconnectness seems to be a weak reason. The related work section lists a number of previous pruning works but it seems difficult to connect them with the proposed approach. On the other hand, related graph-based pruning approaches are missing. For example, [1] proposes to build a graph for each conv layer and prune filters based on the graph complexity.

    [1] Convolutional neural network pruning with structural redundancy reduction. In CVPR 2021.


2. The proposed approach lacks theoretical justification. It is not clear why extracting the longest chain in the graph is the optimal choice for filter pruning.


3. The methodology is not presented well and is hard to follow. It is presented narratively, without formal definitions of the concepts used in the approach and equations, making it difficult to understand the implementation details. Specifically,

    -In Fig. 1(B), what do the blue line, (i) and (ii) mean?

    -It is hard to distinguish the concepts of operation and operator. According to Section 4.1, a node is a channel and an edge is an operator. However, in Fig. 1(B) it seems that a node represents the output of a filter rather than a channel?

    -I think the calculation of operator norms is very important and should be moved from the Appendix to the main text. (On the other hand, the description of datasets in Section 5 is trivial.)

    -It is unclear why $||AB||  \approx ||A||\cdot||B||$.

    -It is unclear how to search for the longest path in the graph.


4. The experiment settings and results are not satisfactory.

    The proposed approach is not evaluated with benchmark classification tasks such as CIFAR and ImageNet. Besides, only structured magnitude and operator norm pruning are used for performance comparison. Both approaches are proposed many years ago. State-of-the-art methods should be added for comparison.

**Summary Of The Paper:**

This paper proposes a structured pruning approach by building a graph according to the structure and weights of the CNN to be pruned. The operators with the longest chain in the graph as preserved while others are pruned. The proposed approach is evaluated with several network structures (MS-D, U-Net, ResNet) and datasets.

**Summary Of The Review:**

Overall, I think there are several nontrivial weaknesses in the manuscript as listed above. I currently give a negative score but I will consider changing it if the authors' feedback resolves the concerns.

---

> ### Author Response · Authors · 2021-11-22
> **Response to Reviewer xFHB**
>
> We thank the reviewer for the useful feedback, which has resulted in several significant improvements in the manuscript. As mentioned in the general update, we agree with the comment made by several reviewers that more comparisons to state-of-the-art pruning methods are necessary. We performed experiments on ResNet50 for all 3 datasets for geometric median pruning (GM) [1], and soft filter pruning (SFP) [2] and have added the results to the manuscript. Furthermore, we have tried to improve the manuscript using the reviewer’s feedback, as follows:
>
> 1. ***The introduction and .. a weak reason.***
>
> The reviewer raises a valid point about theoretical justification. In addition to network disjointness, another motivation is that since convolutions are linear operators, the scaling of an individual convolution within a chain of convolutions is somewhat arbitrary. The original manuscript did not emphasize this motivation for LEAN. We have now added this motivation to the Method section.
>
> 2. ***The related work .. the graph complexity.***
>
> We agree with the reviewer that some of the related works were difficult to connect to the proposed approach, and the manuscript missed references to recent related work. We have updated the related works section with recent references to graph-based pruning methods.
>
> 3. ***The proposed .. filter pruning.***
>
> We address this question in the answer to point 1.
>
> 4. ***The methodology is .. (i) and (ii) mean?***
>
> Based on the feedback we have changed Fig1 on page 4 and we hope it is now clearer. The blue line used to indicate an example longest path in the old figure, it has been removed. (i) and (ii) used to refer to optimizations that can be performed on the pruning graph to reduce the number of nodes and edges. We felt this was unnecessarily complicated for the main point and have created a small section on pruning graph reductions to the Appendix. We hope the new figure clarifies how a CNN can be converted into its pruning graph.
>
> 5. ***It is hard .. than a channel?***
>
> We agree with the reviewer that this was confusing in conjunction with Fig1. First, to clarify, a node is indeed a channel. In the case of a convolutional layer with 1 input channel, a node is therefore indeed the output of a single filter. In the case of multiple input channels it is the output of multiple filters. We think the old Fig1 was confusing because it had single channel inputs, we hope the new Fig1 is clearer in this respect.
>
> 6. ***I think the calculation .. 5 is trivial.***
>
> The reviewer raises a good point. In the revised manuscript, we have moved the mathematical content regarding calculating the operator norms to section 4.3 in the Method section. We have moved some details of the datasets to the Appendix.
>
> 7. ***It is unclear .. approximately $||A|| * ||B||$.***
>
> The reviewer is correct in raising this question, and generally this is indeed not true. We can for example have $||AB||=0$ while $||A||$ and $||B||$ are not zero, e.g., if A and B have orthogonal eigenvectors. $||A||*||B||$ is an upper bound if the submultiplicative property holds. We have expanded this text (Section 4.1) to make clear that multiplying the norms is an upper bound, and that we assume that it is a reasonable approximation even if it may not hold generally. In addition, in the final paragraph of the conclusion on page 9 we mention that stronger approximations of the norm of composed operators are interesting areas for future research to allow LEAN to find more important chains of operators.
>
> 8. ***It is unclear .. path in the graph.***
>
> The reviewer is correct that the explicit graph algorithm was not mentioned. LEAN uses an all-pairs longest path procedure (for DAGs this can be converted into a polynomial time shortest-path algorithm).
> We have mentioned this explicitly in section 4.2 to clarify this. For the algorithmic details we reference Cormen et al..
>
> 5. ***The experiment settings .. added for comparison.***
>
> We understand the reviewers point related to standard classification benchmarks such as ImageNet. In our work we aim to provide significant practical speedup for imaging tasks in real-world applications. These real-world datasets are often small, yet accelerating CNNs has practical benefits in these scenarios. Imaging tasks such as the X-ray CT segmentation task often have very little training data available, for example because they involve manual annotation by domain experts, yet the trained segmentation CNN is applied in real time. Instead of adding ImageNet benchmarks, we think the weaknesses raised by the reviewer are alleviated by comparing to several more advanced methods on our current datasets, which we have included in the revised manuscript. We do aim to benchmark LEAN on the ImageNet database in the future.
>
> [1] Filter Pruning via Geometric Median for Deep Convolutional Neural Networks Acceleration, CVPR 2019
>
> [2] Soft Filter Pruning for Accelerating Deep Convolutional Neural Networks, IJCAI 2018

---

> > ### Comment · Reviewer_xFHB · 2021-11-29
> > **post-rebuttal comments**
> >
> > First of all, I would like to thank the authors for the feedback. Some of my previous concerns are resolved and the updated manuscript looks much better. There are still some issues that could be further improved.
> >
> > (1) I still feel that the theoretical justification of the proposed approach is not fully convincing. Although the experiment results show improvements, I'm still not fully convinced why it works well.
> >
> > (2) The authors added the comparison with some recent work [1-2], but more recent works (say, those in 2020 and 2021) are still missing.
> >
> > (3) I still think that experiments with benchmark datasets such as ImageNet is needed to validate the effectiveness of the proposed approach.
> >
> > Overall, I think the quality of the manuscript is improved. But there are still some issues that can be improved. Based on these concerns, I decide to increase my rating to 5.
> >
> > [1] Filter Pruning via Geometric Median for Deep Convolutional Neural Networks Acceleration, CVPR 2019
> > [2] Soft Filter Pruning for Accelerating Deep Convolutional Neural Networks, IJCAI 2018

---

### Official Review · Reviewer_wyEq · 2021-11-02

**Correctness:** 3
**Technical Novelty And Significance:** 3
**Empirical Novelty And Significance:** 3
**Recommendation:** 5
**Confidence:** 3

**Main Review:**

Strengths:
Propose the new concept of the longest chain path over DAG, which is used in model pruning.

Weakness:
1.  The algorithm is actually in a greedy manner to remove the path one by one. Actually, in previous existing work, there are some discussions about how to deal with the wrongly pruned channels at the early stages of pruning. Any discussion about this issue?
2.  Any comparisons on classification benchmarks on imagenet? I think as a model pruning method, it is necessary to compare with other many SOTA pruning methods on classification benchmarks, to demonstrate its superiority.
3. For image segmentation tasks mainly discussed in the paper, the comparing methods are the only magnitude and operator norm, which I think is also not enough.

**Summary Of The Paper:**

This paper abstracts the AI model as DAG and proposes to use the longest chain path with accumulated operator norm as criteria to perform a greedy pruning method.

**Summary Of The Review:**

Due to the aforementioned weakness, I temporarily think this paper is marginally below the acceptance threshold.

---

> ### Author Response · Authors · 2021-11-22
> **Response to Reviewer wyEq**
>
> We thank the reviewer for the useful feedback, which has resulted in several significant improvements in the manuscript. As mentioned in the general update, we agree with the comment made by several reviewers that more comparisons to state-of-the-art pruning methods are necessary. We performed experiments on ResNet50 for all 3 datasets for geometric median pruning (GM) [1], and soft filter pruning (SFP) [2] and have added the results to the manuscript. Furthermore, we have tried to improve the manuscript using the reviewer’s feedback, as follows:
>
> 1. ***The algorithm is actually in a greedy manner to remove the path one by one. Actually, in previous existing work, there are some discussions about how to deal with the wrongly pruned channels at the early stages of pruning. Any discussion about this issue?***
>
> The reviewer raises an interesting point. We are aware of the work by Guo et al [3] on connection splicing to avoid incorrect pruning where instead of a greedy approach, an optimization problem is solved.
>
> In the second paragraph of the conclusion on page 9, we now discuss that indeed the approach by Guo et al could be used to prevent incorrect pruning due to greediness.
>
> 2. ***Any comparisons on classification benchmarks on imagenet? I think as a model pruning method, it is necessary to compare with other many SOTA pruning methods on classification benchmarks, to demonstrate its superiority. For image segmentation tasks mainly discussed in the paper, the comparing methods are the only magnitude and operator norm, which I think is also not enough.***
>
> We understand the reviewers point related to standard classification benchmarks such as ImageNet. In our work we aim to provide significant practical speedup for imaging tasks in real-world applications. These real-world datasets are often small, yet accelerating CNNs has practical benefits in these scenarios. Imaging tasks such as the X-ray CT segmentation task often have very little training data available, for example because they involve manual annotation by domain experts, yet the trained segmentation CNN is applied in real time. Instead of adding ImageNet benchmarks, we think the weaknesses raised by the reviewer are alleviated by comparing to several more advanced methods on our current datasets, which we have included in the revised manuscript. We do aim to benchmark LEAN on the ImageNet database in the future.
>
> We would once again like to thank the reviewer for the feedback. We hope we have answered your questions.
>
> [1] Filter Pruning via Geometric Median for Deep Convolutional Neural Networks Acceleration, CVPR 2019
>
> [2] Soft Filter Pruning for Accelerating Deep Convolutional Neural Networks, IJCAI 2018
>
> [3] Dynamic Network Surgery for Efficient DNNs, Guo et al., NeurIPS 2016.

---

> > ### Comment · Reviewer_wyEq · 2021-11-29
> > **post-rebuttal comments**
> >
> > Thank you for your response.
> > I think the content of this paper is trying to claim that it proposes a new pruning method, however, the paper only provides comparisons on several small datasets, instead of the most decent benchmark dataset ImageNet.  I cannot buy it this way.  The comparisons with several NON-sota methods on those small datasets are also not convincing.
> > Thus, I will not increase my score.

---

### Official Review · Reviewer_rCV2 · 2021-11-08

**Correctness:** 3
**Technical Novelty And Significance:** 2
**Empirical Novelty And Significance:** 2
**Recommendation:** 3
**Confidence:** 4

**Main Review:**

Strengths:
1. The idea of using a graph-based algorithm for structured pruning is novel and interesting. Since LEAN prunes the network by keeping the longest path in the graph, it can use the global pruning ratio without worrying about generating a disconnected network. Therefore, it may prune much more aggressively in some layers than the other two structured pruning methods.
2. Experiment results show great potentials for LEAN pruning.

Weaknesses:
1. A primary concern is on the evaluation. (a) The paper only compares LEAN with two naive structured pruning strategies, but not other more advanced methods. (b) The datasets used are too small (e.g., 1000 training examples in the Simulated CS dataset). It is not clear whether applying ResNet50 or U-Net4 is reasonable. Why not use those commonly used datasets (e.g., Cifar10, ImageNet) and networks for evaluation? (c) The speedup results are only shown for an MS-D network on the dynamic X-ray CT dataset. And the paper mentioned that it uses a custom MS-D model which loads only the unpruned filters. Then I wonder what the speedups of networks pruned with other structured pruning methods or unstructured pruning methods are with such customized implementations.
2. I would like to see some theoretical analysis on why LEAN outperforms the basic structured pruning methods such as structured magnitude pruning or operator norm pruning. It may help make the paper more rigorous.
3. The authors may want to improve the presentation to better describe the definition of preliminaries and how to construct the pruning graph. Figure 1 (B) is confusing.


Minor comments:
1. The figures are not black/white printer friendly.
2. Most of the related works discussed in section 2 are those before 2020. I would like to see more discussions on the most recent related pruning works.
3. It might be better also to report the accuracy/F1-score/mIoU of the unpruned network in the paper.
4. It would be good to see the structure of the pruned networks with LEAN in either the main paper or the appendix.
5. In the last paragraph of section 6.1, the paper mentions structured magnitude pruning performs slightly better than LEAN for a few pruning ratios with U-Net4 and the CamVid dataset. But it doesn't give any explanation or discussion.

**Summary Of The Paper:**

This paper proposes the LongEst-chAiN (LEAN) method to perform structured pruning of CNN networks. LEAN maps a CNN network to a pruning graph, where every channel of input/output is a node, and every operator is an edge connecting input and output nodes. It uses the operator norms as the weights of edges. Then it prunes the network by keeping the longest path in the graph iteratively until it reaches the target pruning ratio. This paper demonstrates the effectiveness of LEAN pruning by comparing it to two structured pruning methods (structured magnitude pruning, operator norm pruning) across three image segmentation datasets (Simulated Circle-Square dataset, CamVid, Real-world dynamic CT dataset) and three CNN architectures (MS-D, U-Net4, and ResNet50). Experiment results show that LEAN outperforms the other two structured pruning methods as it achieves similar model qualities with much smaller pruning ratios in most cases. Also, the paper shows that a MS-D model pruned with LEAN is 10.9X faster than the unpruned network in practice.

**Summary Of The Review:**

Overall, the idea of the paper is novel and interesting, and the experiment results look promising. However, the paper fails to compare LEAN with the most advanced related works on structured pruning. The datasets used in the experiments are too small, and the experimented tasks are not commonly used in CNN pruning works. So it is hard to justify the advances of the proposed pruning method. Also, the presentation of the paper needs to be improved.

---

> ### Author Response · Authors · 2021-11-22
> **Response to Reviewer rCV2**
>
> We thank the reviewer for the useful feedback, which has resulted in several significant improvements in the manuscript:
>
> 1. ***A primary concern .. more advanced methods.***
>
> We agree with the reviewer’s remarks and we added two experimental comparisons to more advanced methods. We have added experimental results for GM [1] and SFP [2] for ResNet50 for all datasets.
>
> 2. ***(b) The datasets used .. networks for evaluation?***
>
> Regarding the reasonableness of applying ResNet and UNet to the CS and X-ray datasets: for these datasets, the unpruned metrics of UNet and ResNet are similar to the metrics of MS-D. Overfitting therefore seems to be a minor issue. We see that these networks can be pruned quite aggressively on these tasks using LEAN, but not by as much using other pruning methods. In many cases, we see these types of networks (especially UNet) being applied to similar data in practice.
>
> We understand the reviewers point related to standard classification benchmarks such as ImageNet. In our work we aim to provide significant practical speedup for imaging tasks in real-world applications. These real-world datasets are often small, yet accelerating CNNs has practical benefits in these scenarios. Imaging tasks such as the X-ray CT segmentation task often have very little training data available, for example because they involve manual annotation by domain experts, yet the trained segmentation CNN is applied in real time. Instead of adding ImageNet benchmarks, we think the weaknesses raised by the reviewer are alleviated by comparing to several more advanced methods on our current datasets, which we have included in the revised manuscript. We do aim to benchmark LEAN on the ImageNet database in the future.
>
> 3. ***(c) The speedup .. customized implementations.***
>
> The reviewer raises a good point. We have included the practical speedup for structured operator norm pruning (which was the best performing method other than LEAN) in Fig 4.
>
> 4. ***I would like .. paper more rigorous.***
>
> The reviewer raises a valid point about theoretical justification. In addition to network disjointness, another motivation is that since convolutions are linear operators, the scaling of an individual convolution within a chain of convolutions is somewhat arbitrary. The original manuscript did not emphasize this motivation for LEAN. We have now added this motivation to the Method section.
>
> 5. ***The authors may .. Fig 1(B) is confusing.***
>
> We have added additional structure to the Method section about how to construct the pruning graph. Furthermore, we changed Fig1 to display the full pruning graph (so every operator is included) rather than the “optimized” pruning graph where we performed reductions to reduce the number of edges and nodes.  We decided that these reductions are not of main importance to the paper, and have moved these to the Appendix.
>
> 6. ***The figures are not black/white printer friendly.***
>
> We have added markers to the graphs in Figs 3 & 4 and changed line styles to make them black/white friendly.
>
> 7. ***Most of the related .. related pruning works.***
>
> We have updated the Related Work section and added several references to more recent pruning works.
>
> 8. ***It might be .. network in the paper.***
>
> In the original manuscript, the first data point in the upper right corner of each graph in Fig3 is the performance of the unpruned network. Reading our original manuscript again, we now realize that this was not clearly indicated. To emphasize this we have added a red circle in the graphs in Fig3. The unpruned accuracies are also given in section 5.3 pg 7.
>
> 9. ***It would be .. or the appendix.***
>
> We plotted the adjacency matrices of pruned networks where a value is 0 if it is pruned (black) and 1 if it is still active (white). The adjacency matrices are directly related to the graph structure and show some interesting differences between the pruning methods. Because the MS-D net has only =-5k convolutions, and is densely connection, we can plot the full adjacency matrix. We have added Fig6 in the Appendix that shows these matrices of pruned MS-D networks.
>
> 10. ***In the last .. or discussion.***
>
> Indeed, a possible explanation for the relative performance between magnitude pruning and LEAN in this case was lacking in the original manuscript. Due to the large changes (already after only a few pruning steps) between different runs of the same algorithm, we believe that 5 epochs of retraining is perhaps insufficient for CamVid to fully recover accuracy. In this phase the pruning methods appear to perform similarly. We think that the fact that LEAN’s average lies below structured magnitude pruning for two pruning ratios is likely not statistically significant. We have added this possible explanation in section 6.1 pg 9.
>
> [1] Filter Pruning via Geometric Median for Deep Convolutional Neural Networks Acceleration, CVPR 2019
>
> [2] Soft Filter Pruning for Accelerating Deep Convolutional Neural Networks, IJCAI 2018

---

### Official Review · Reviewer_XPch · 2021-11-09

**Correctness:** 3
**Technical Novelty And Significance:** 3
**Empirical Novelty And Significance:** 3
**Recommendation:** 5
**Confidence:** 3

**Details Of Ethics Concerns:**

None.

**Main Review:**

The method is simple in its approach, which allows the method to be easy to understand.  Suggestions for improvement are minor: Comparisons to additional methods presented in the related work and corresponding benchmarks would strengthen the results (e.g. ResNet per Dong & Yang 2019). Benchmarks for the full network could help better contextualize pruned results. Replicates are presented in a way that is difficult to distinguish methods from one another.  For example it's difficult to distinguish light blue vs gray lines in Figure 3.  Additionally, variability of speedup between training runs is not presented in figure 4. Moreover, the relative number of remaining convolutions metric is presented a bit vaguely.  It's not clear why some panels have these metrics missing.  Additionally, it's not clear how one would calculate what accuracy is used to calculate "relative number of convolutions at equal accuracy" for each graph.

**Summary Of The Paper:**

The authors propose a structured pruning method which turns structure pruning into a graph pruning problem.  The authors represent each input, output pair as an operator node, with edges measured by operator norms between operators.  The authors then propose an iterative structured pruning algorithm which prunes layers based on the longest path in the graph.  They then evaluate their method on three image segmentation tasks.

**Summary Of The Review:**

Overall, the method is easy to understand and the paper is clearly presented.  Some organizational and presentation changes could help make the results easier to interpret.

---

> ### Author Response · Authors · 2021-11-22
> **Response to Reviewer XPch**
>
> We thank the reviewer for the useful feedback, which has resulted in several significant improvements in the manuscript. As mentioned in the general update, we agree with the comment made by several reviewers that more comparisons to state-of-the-art pruning methods are necessary. We performed experiments on ResNet50 for all 3 datasets for geometric median pruning (GM) [1], and soft filter pruning (SFP) [2] and have added the results to the manuscript. Furthermore, we have tried to improve the manuscript using the reviewer’s feedback, as follows:
>
> 1. ***Comparisons to additional methods presented in the related work and corresponding benchmarks would strengthen the results (e.g. esNet per Dong & Yang 2019).***
>
> We agree with the reviewer’s remarks and we added two experimental comparisons to more advanced methods. We have added experimental results for GM [1] and SFP [2] for ResNet50 for all datasets.
>
> 2. ***Benchmarks for the full network could help better contextualize pruned results.***
>
> In the original manuscript, the first data point in the upper right corner of each graph in Figure 3 on page 8 is the performance of the unpruned network. Reading our original manuscript again, we now realize that this was not clearly indicated.
> To emphasize that the unpruned network performance is shown, we have added a red circle in the graphs in Figure 3. The unpruned accuracies are also given in section 5.3 on page 7.
>
> 3. ***Replicates are presented in a way that is difficult to distinguish methods from one another. For example it's difficult to distinguish light blue vs gray lines in Figure 3.***
>
> To make the graphs better distinguishable, we have changed the line styles and added markers for different methods in Figure 3 (page 8) and Figure 4 (page 9).
>
> 4. ***Additionally, variability of speedup between training runs is not presented in figure 4.***
>
> We have added the standard deviation over 5 timing runs in Figure 4 (page 9). However, the effect is so small that the error bars are barely visible.
>
> 5. ***Moreover, the relative number of remaining convolutions metric is presented a bit vaguely. It's not clear why some panels have these metrics missing.***
>
> We have added a formula in the text (see the second paragraph of section 5 on page 6) to clarify this metric. In Figure 3 (page 8) we display the x and y-axis labels on the sides and at the bottom, but they hold for all panels. If the reviewer finds this confusing, we can add axis labels for all panels in Figure 3, but they were initially left out to make the Figure less cluttered.
>
> 6. ***Additionally, it's not clear how one would calculate what accuracy is used to calculate "relative number of convolutions at equal accuracy" for each graph.***
>
> We assume the reviewer is referring to the orange arrows that indicated the speedup with LEAN in Figure 3. To clarify, the drop-off accuracies were initially estimated by hand. We agree that it was unclear how this accuracy was chosen. In the revised manuscript, for ResNet50 we now measure the relative number of convolutions at equal accuracy by looking at the best pruned networks with a 3% accuracy reduction (or less) compared to the unpruned networks. This is described in section 6.1 on page 8. In addition, since we now compare with more methods, we decided that the arrows created a cluttered figure and have removed them altogether from Figure 3.
>
> We would once again like to thank the reviewer for the feedback. We hope we have answered your questions.
>
> [1] Filter Pruning via Geometric Median for Deep Convolutional Neural Networks Acceleration, CVPR 2019
>
> [2] Soft Filter Pruning for Accelerating Deep Convolutional Neural Networks, IJCAI 2018

---

### Author Response · Authors · 2021-11-16
**Update: Adding SOTA experimental comparisons**

Dear reviewers and area chair,

We thank the reviewers for their feedback and we are working on incorporating the comments. Here, we already want to post an update on our efforts to obtain new experimental results. We agree with the reviewers that more comparisons to SOTA pruning methods are necessary. We are currently working on incorporating several SOTA comparisons. After evaluating the published code of many comparison methods, we aim to at least include experimental results for two methods from Table 2 in [1] (perhaps 3 methods). Due to time constraints, we limit ourselves to methods that already implement algorithms to prune ResNet50.

Although comparisons using ImageNet data would be informative, for the current manuscript we choose to leave such comparisons for future work. We make this choice as we ultimately aim, in this manuscript, to provide significant practical speedup for image segmentation tasks in real-world applications. In practice, such tasks often have very little training data available, for example because they involve manual annotation by domain experts. Therefore, comparisons using tasks with limited amounts of training data are highly relevant for such use-cases. Moreover, the ImageNet database is very large, and we do not have the computational facilities to perform ImageNet pruning experiments in the limited time available for revision. We do aim to benchmark LEAN on the ImageNet database in the future.

We hope that our efforts strengthen the value of LEAN in the eyes of the reviewers. We appreciate any input from reviewers should they wish to comment on our efforts to strengthen the experimental results, or offer suggestions. We will also be addressing the comments on the manuscript soon.

Kind regards,

Authors

[1] Convolutional neural network pruning with structural redundancy reduction, CVPR 2021

---

### Decision · Program_Chairs · 2022-01-20

**Decision:**

Reject

**Comment:**

I do not recommend accepting this paper, although I make this decision with reservations. The review quality for this paper was not particularly strong, and I wish to emphasize to the authors that I read the paper myself in detail in the process of writing this metareview.

This paper proposes a new structured pruning technique called LEAN. It involves computing an operator norm of the convolutions in a convolutional neural network, multiplying these norms over paths through the network, and keeping the paths with the highest such values (and pruning everything else). This paper makes the argument that this metric is robust to scaling and prevents network discontinuities. (In this way, the technique is very reminiscent of SynFlow (Tanaka et al., Pruning Neural Networks without any data by iteratively conserving synaptic flow) in terms of motivation and resulting technique, although SynFlow is unstructured. I do not mean this as a criticism - just a suggestion for the authors of a connection they might be able to make in the future.)

One big concern I have about this paper based on the methodology alone is as follows: the paper states a number of hypotheses about why this is a sensible way to prune (e.g., in the beginning of Section 4). I see no reason why any of these hypotheses are wrong, but the paper never makes an effort to evaluate any of them. I don't mean a theoretical justification here - that's difficult and unlikely to yield useful information about what happens in practice. I mean experiments to ablate the salient properties of the heuristic mentioned in the paragraph at the beginning of Section 4 (Does scaling invariance actually matter in practice? Is network disconnectivity actually a risk in practice?).

My biggest concerns about the paper, however, are in the evaluation. I share two major reviewer concerns that were mentioned:
(1) The paper compares to a very limited set of baseline pruning methods, and relatively older ones at that (2019 is indeed old in the world of pruning).
(2) The paper does not look at standard, "large-scale" benchmarks for computer vision - namely, ResNet-50 on ImageNet.

Neither of these concerns is necessarily decisive in my view. For example, with respect to Concern 1, the reviewers unhelpfully do not suggest very many additional structured pruning benchmarks to consider, and I think the additional baselines added during the revision process have softened this concern. I would also recommend taking a look at "Growing Efficient Deep Networks by Structured Sparsification" (Yuan et al) for a useful method and a good set of baselines. There are an arbitrary number of baselines one could add and the structured pruning space is a confusing mess, but I think the claims in this paper merit more than are currently present.

With respect to Concern 2, I'm even more conflicted. On the one hand, I have rarely seen any pruning techniques proposed for or evaluated on vision tasks beyond image classification, despite the fact that - in the real world - segmentation is much more popular than it would seem by reading the ICLR proceedings. To that end, I applaud the authors for focusing on those settings and I see substantial value in a paper that does so. On the other hand, ResNet-50 on ImageNet (among other standard classification benchmarks) is the de facto measuring stick for evaluating pruning methods in computer vision, and the exclusive focus on segmentation here means it is very difficult to compare the proposed technique to other benchmarks. If the paper is to focus on segmentation alone, this places a higher burden on adding many additional comparisons to other methods (i.e., Concern 1). Finally, I don't see any reason why the paper *couldn't* also include ResNet-50 on ImageNet or the like in addition to segmentation; if it is a limitation on the compute available to the authors (something I empathize with), they did not say so in any of the author responses. Upon reading the author responses, I was left asking, "Why not both?"

For those reasons, I do not recommend accepting the paper, although I think there are some good reasons to value the paper's contributions. At the end of the day, there are some relatively simple things that could be changed to make the paper much easier to contextualize within the pruning literature. As of right now, it would be very difficult for me to say whether or in what contexts this method should actually be used in practice.

(P.S. I agree with the reviewers that Figure 3 is exceptionally hard to parse.)